# Requirements and Solution Approaches to Personality-Adaptive Conversational Agents in Mental Health Care

**Dominik Siemon** [1] 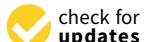**, Rangina Ahmad** [2,*]**, Henrik Harms** [2] **and Triparna de Vreede** [3]

1   Department of Software Engineering, School of Engineering Science, LUT University,
    53850 Lappeenranta, Finland; dominik.siemon@lut.fi
2   Chair of Information Management, Institute of Business Information Systems, Technische Universität
    Braunschweig, 38106 Braunschweig, Germany; henrik.harms@tu-braunschweig.de
3   School of Information Systems and Management, Muma College of Business, University of South Florida,
    Tampa, FL 33620, USA; tdevreede@usf.edu
*   Correspondence: rangina.ahmad@tu-braunschweig.de; Tel.: +49-531-391-3139

**Abstract:** Artificial intelligence (AI) technologies enable Conversational Agents (CAs) to perform highly complex tasks in a human-like manner and may help people cope with anxiety to improve their mental health and well-being. To support patients with their mental well-being in an authentic way, CAs need to be imbued with human-like behavior, such as personality. In this paper we cover an innovative form of CA, so-called Personality-Adaptive Conversational Agents (PACAs) that automatically infer users' personality traits and adapt accordingly to their personality. We empirically investigate their benefits and caveats in mental health care. The results of our study show that PACAs can be beneficial for mental health support, but they also raise concerns about trust and privacy issues. We present a set of relevant requirements for designing PACAs and provide solution approaches that can be followed when designing and implementing PACAs for mental health care.

**Keywords:** conversational agents; Personality-Adaptive Conversational Agents; mental health care

## 1. Introduction

One of the striking aspects of the current state of global health is the ever-rising need for therapeutic and mental health care. For instance, the number of sick leaves due to mental illness in Germany has more than doubled since 1997 [1]. While this crisis is global in nature, its effects are more apparent in countries with weaker health care systems. For example, a whopping 77% of global mental-health related suicides occur in low- or middle-income countries [2]. Despite the increasing need for mental health care, the number of healthcare workers remains incredibly low in these countries. It is estimated that the supply of healthcare workers in low-income countries is as low as 2 per 100,000 inhabitants while high-income countries have an average of 70 health care workers per 100,000 habitants [3]. The COVID-19 pandemic further highlighted the urgent need for additional investments in the mental health care sector. In the first year of the pandemic alone, the emergency calls related to mental health conditions, suicide attempts, drug overdose, or child abuse increased significantly in the United States [4]. The negative effects of long-lasting lockdowns can also be observed globally, with people often suffering from loneliness or depression [2].

Given the increasing need for mental health care workers coupled with less than adequate supply in the short- to mid-term time frame, alternatives to mental health care need to be considered to prevent an all-out mental health crisis. Empathetic Artificial Intelligence (AI) agents or Conversational Agents (CAs) [5] have emerged in the recent past as a viable alternative because they are accessible anywhere and available at any time to provide counseling and deliver therapeutic interventions [3]. CAs may help people cope

with mental health conditions such as depressive anxiety and loneliness to enhance mental health and well-being [6].

While the idea of CAs mitigating the mental health crisis is an appealing one, it would be unwise to implement it on a large scale without evaluating the impact of such CA based therapy on individuals and societies. This conversation regarding the benefits and dark sides of AI is not new. While the concept of CAs for mental health has been popularized recently, the attempts to create "therapy bots" have been in progress for decades [7]. It began in 1966, when computer scientist Joseph Weizenbaum created an empathetic machine to simulate a psychotherapist. The technology behind the "empathetic" machine was a simple computer program called ELIZA that was able to communicate with humans via natural language [8]. He witnessed how his participants opened their hearts to a computer program and was shocked by their emotional attachment to the program. This experience turned him into an ardent critic of his own creation [9]. While the people interacting with the machine ascribed human characteristics to it and other psychiatrists saw ELIZA's potential as a computer-based therapy and as a "form of psychological treatment" [10] (p. 305), Weizenbaum himself had misgivings about this mode of therapy. He wanted to "rob ELIZA of the aura of magic to which its application to psychological subject matter has to some extent contributed" [8] (p. 43).

Fast forward to almost 6 decades later, and the debate still rages on. While there are an increasing number of therapy bots in the market, critics continue to advocate for the need to reassess the potential "dark sides" of AI and the ethical responsibilities of developers and designers [11,12]. We now live in a world where ELIZA's descendants have become an integral part of people's lives and have names such as Woebot and Replika. They have evolved into "digital creatures that express human-like feelings" [11] (p. 1) and have become increasingly capable of handling highly complex tasks with human qualities such as a higher autonomy of decision-making [12,13].

The domain of mental health care is a highly patient-centered sphere, where a successful conversation is dependent on patients' individual dynamic behavior and the therapist's ability to adapt to the patient's specific needs in order to form a therapeutic relationship [14,15]. As the capabilities of CAs continue to evolve, they are able to personalize the mental health care [14] by capturing their individual dynamic behavior and adapting to the users' specific personalities. One of the goals of this paper is to examine such a specific type of CA—a Personality-Adaptive Conversational Agent (PACA). PACA represents a novel way to address a serious prevalent problem (mental health issues). The concept of a PACA is an innovative solution that uses AI to demonstrate progress in the field of healthcare [16–18]. PACAs automatically infer users' personality traits and adapt to their personality by using language that is specific to a particular personality dimension (e.g., extraversion, agreeableness) to enhance dialogue quality [19,20]. Thus, PACAs establish rapport with the patient to enhance interaction quality and mental health support.

While we celebrate the progress of AI assisted mental health care agents, it is also important to highlight the caveats of creating and perfecting human-like CAs with simulated feelings, without considering long-term consequences for human beings such as deep, emotional attachments as was demonstrated by ELIZA [8,11]. Accordingly, determining the degree of likeness to humans [11] poses a challenge for CA designers. Designing CAs that are capable of expressing human-like characteristics such as a personality and yet withholding from them the expressions of feelings and empathy because they "are the very substance of our humanness" [11] (p. 1) is a matter of delicate balance. This dilemma is brought to sharper focus with the increasingly acute shortage of mental health workforce globally making empathetic CAs a promising source of support [2,3,6].

In light of the caveats that need to be considered with human-like CAs that do not have "real" feelings, it is critical to take ethical issues (i.e., trust, privacy, support) into consideration when designing PACAs. In addition, it is necessary to identify the apprehensions people have against the use of PACAs and furthermore, how the design of PACAs

can support overcoming these caveats. Consequently, in this paper we focus on answering the following research questions:

RQ1: What are benefits of PACAs in mental health care?

RQ2: What caveats do we need to be aware of regarding the usage of PACAs in mental health care?

RQ3: Which requirements could be derived from the identified caveats and what are the solutions to address the requirements?

To address these research questions, we followed an explorative research approach and conducted a qualitative study [21]. The results of this study contribute to understanding of PACAs' overlooked benefits and the emerging caveats in mental health care, particularly the potential positive and negative aspects of PACAs. Furthermore, we specifically focus on the caveats and recommend solutions to address them.

The remainder of this paper is structured as follows: In the theoretical background, we first give a brief overview of selected CAs used in a mental health care context and elaborate on the concept and functionalities of a PACA. We then explain our method and how we simulated a conversation between a PACA therapist and a human patient to conduct our qualitative study. We then present the results and discuss our RQ.

## 2. Theoretical Background

### 2.1. State of the Global Mental Health Care

Unfortunately, in the recent years, global state of mental health care has not significantly changed for the better. According to the WHO, even economically strong nations have not met their goals jointly set with the WHO in 2014 [2]. A recent analysis of over 190 million emergency department visits in the United States found that emergency calls related to mental health conditions, suicide attempts, drug overdose, or child abuse have substantially increased in the period from March 2020 to October 2020, compared to the same period in 2019 [4]. The authors strongly believe that this development is correlated to the COVID-19 pandemic and the ensuing lockdown. Lockdowns have resulted in a rising number of mental health cases across the world. Not only did lockdowns result in new mental health challenges, but they also revealed the underlying issues that were causing the mental health conditions [2].

Individuals who were isolated and lonely were not the only victims to the mental health challenges; there were other people such as the front-line workers in the public health care system who were also deeply affected. This is especially true for heavily affected regions, such as Lombardy in Italy, where mental health care became, and still is, a focus topic after the first wave of drastically high numbers of COVID-19 infections and mortality rates [2]. Incidents like these should be monitored closely and be taken into account by decision-makers, as more health care providers struggle to meet the demand [4].

To counter the effects of this crisis, large investments are needed for the mental health sector. Tragically, despite the urgency, most of the overall mental health goals set by the WHO were not reached by 2020. According to the WHO, only 52% of all 194 member states met the targets for mental health promotion and prevention programs [2]. For majority of the countries, budget for mental health resources did not significantly increase and remains at around 2% of the total health budget. Even countries, whose plans included estimations on the increasing need for financial and human resources have only managed to provide these resources in 39% of the cases [2].

One of the major goals of the WHO has been the decentralization of mental health care from large, centralized institutions towards integration into primary health care in the communities. In middle-income countries, around 70% of the government's mental health budget is allocated to central mental health hospitals, compared to an average of 35% in high-income countries [2]. While the accessibility for mental health care improves with decentralization, people in middle-income countries still have higher entry barriers for mental health care. Countries need to overcome this accessibility challenge to prevent a further increase in mental health conditions. A potential solution could be the use of CAs

that interact with people through computers and mobile apps. As over 80% of the world's population own a smartphone, the technology could potentially be easy to access for the vast majority of people [22].

### 2.2. Conversational Agents in Mental-Health Care

CAs are software-based systems designed to interact with humans using natural language [23]. One emerging area in which conversational technologies have the potential to enhance positive outcomes is in mental healthcare [24]. ELIZA is widely considered the first functional CA in history [9]. While it took on the role of a Rogerian therapist, the program appeared in a psychotherapeutic context for demonstration purposes only and was not made available to the public [8]. The underlying technology of ELIZA was simple: By searching the textual input of its conversation partner for relevant keywords, the machine produced appropriate responses according to rules and directions based on hand-crafted scripts by the programmers [9]. PARRY, another early example of a prototype CA developed in 1979, was a contrast to ELIZA. It was designed to simulate and behave like a person with paranoid schizophrenia [25]. The developers' intention was to find out if other psychiatrists could determine a real paranoid patient from their computer model [26]. According to Heiser et al. (1979, p. 159) their approach was not only "valuable to researchers in computer science and psychopathology" but also helpful for mental health educators [27]. While PARRY passed the Turing test for the first time in history, it was still a rule-based CA and had a similar functioning as ELIZA, though with better language understanding capabilities [25].

Currently, CAs are developed based on advanced AI technologies such as machine learning and natural language processing and have more powerful capabilities to support mental health care, well-being, or sustainable activities. They are also more widely available to the public and are being applied to various domains. Their application varies from allowing "people with intellectual disabilities to work and train their social skills" [13] (p. 1) or to support individuals make more sustainable financial decisions by saving transaction and management costs [28], just to name a few. Furthermore, there are commercially available mobile phone CA applications that help people manage symptoms of anxiety and depression by teaching them self-care and mindfulness techniques [3]. One such application, Woebot, is a CA engineered to assess, monitor, and respond to users dealing with mental health issues [29]. Woebot provides a responsive and adaptive intervention to its users with in-the-moment help and targeted therapies aligned to their evolving symptoms and needs [30]. According to its developers, Woebot is a CA that builds a bond with its users by trying to motivate and engage them in a conversation about their mental health [30]. Replika is another commercially available CA app that pursues a similar approach as Woebot. It presents itself to the user as "an AI companion who cares" [6,31]. Its users report that it is great to have someone to talk to who does not "judge you when you have, for example, anxiety attacks and a lot of stress" [31]. Replika is built to closely resemble natural human communication and it learns more about the user from their interactions making it more personable over time [31,32]. However, neither Woebot nor Replika automatically infer personality traits and neither do they adapt a personality according to the user's need. To the best of our knowledge, the concept of a personality-adaptive CA does not exist in the marketplace.

### 2.3. Personality-Adaptive Conversational Agents

Contemporary CAs, such as Woebot and Replika, are designed to reflect specific human characteristics that are appealing to the user. These characteristics have been given various names such as "human-like behavior", "anthropomorphic features", or "social cues" [23] and they have a consistent positive impact on user interaction quality and user experience [33,34]. Along with other anthropomorphic characteristics such as gender, voice, and facial expressions, personality has been identified as one of the key components of CAs designed for long-term conversations [23,35,36]. Psychologists have

documented the existence of personality cues in language across a wide range of linguistic levels by discovering correlations between a number of linguistic variables and personality traits [37]. The more frequently a person exhibits a trait in their language markers, the more consistently that trait will be a factor in their behavior [37]. Of the Big Five personality dimensions, two traits appear to be particularly meaningful in the context of interpersonal interaction: agreeableness and extraversion [38,39]. For example, extraverts are likely to have higher rate of speech, speak more, louder, and more repeatedly, with fewer hesitations and pauses, shorter silences, higher verbal output, and less formal language. People who are highly agreeable show a lot of empathy, agree, compliment, use longer words and many insight words, and make fewer personal attacks towards their conversation partner [37]. If such personality cues are categorized and coded, then developers who seek to design CAs that display typical characteristics of a human therapist, such as caring and empathetic communication, can select a relevant set of personality cues to make the CA appear empathetic. If communication is through a chat box, then the caring personality can be programmed to manifest using a natural language. These language cues induce a specific personality to a CA in their conversational design.

Even though a lot of research exists on how to endow machines with personality [19,20,35,36] and how to adapt conversation style, most contemporary CAs are still focusing on a "one-size-fits-all"-design. This means that once a CA is developed with a specific personality for a certain domain, the CA is not capable of changing its personality when interacting with individual users. However, since the needs of users can be fundamentally different, CAs is more effective if they are user and personality-adaptive in to be able to accommodate to different user needs. Due to advances in technology, specifically language-based personality mining tools such as IBM Watson's Personality Insights, LIWC, GloVe, or 3-Grams, it is now possible to design a PACA in a manner that is capable of automatically inferring personality traits from an individual's speech or text [40]. Natural language processing techniques are particularly attuned to the interpretation of massive volumes of natural language elements by recognizing grammatical rules (e.g., syntax, context, usage patterns) of a word, sentence, or document [40]. In practice, a PACA would analyze users' text data such as their chat histories or social media posts to derive users' personality traits. Once the (dominant) traits are identified, the PACA adapts and responds accordingly in written or spoken language [20]. Used in a mental healthcare context, a PACA's main task would be to attune its personality to best able to socially support patients in stressful situations. However, since in this specific domain both data and patients are particularly sensitive, it is crucial to know what potential trust and privacy concerns a PACA might deal with—especially with regard to the degree of human-likeness of CAs and the potential dangers of humans becoming emotionally too attached to the machine because of the high personality fit between the two.

## 3. Methodology

### 3.1. Sample and Data Collection Procedure

To address our research questions, we followed an explorative research approach and conducted a qualitative study [21], constructing an open questionnaire with the aim to capture comprehensive opinions about PACAs in the context of mental health care regarding support, trust, and privacy. Our open questionnaire consisted of an extensive explanation of the functionality and nature of a PACA to make sure participants from all backgrounds understood the concept. The description of what a PACA is, how it works, and where it can be used was explained in detail (written description). In addition, an example of a simulated conversation between a PACA therapist and a human patient was provided to the participants (see section PACA Design). Next, we used several open questions to cover the categories of support, trust, and privacy asking for input by the participants. The following Table 1 provides an overview of these open questions.

**Table 1.** Open Questions for the Categories Support, Trust, Privacy.

| Category | Question (Asked to Explain in at Least 2–3 Sentences) |
|---|---|
| Support | Do you think the concept of a PACA is useful/ helpful in mental health therapy? |
| | What are the reasons that speak against communicating with a PACA? What concerns would you have in your interaction with the PACA? |
| | Could a PACA pose a danger? Would you be afraid that the PACA could become manipulative and tell you things that would be rather counterproductive for your mental health? |
| Trust | Would you trust a PACA and can you imagine building a relationship with the PACA over a longer period of time? Would you also maintain the relationship with a PACA? |
| Privacy | Would you agree to give the PACA access to your data? Would you have privacy concerns? |

The survey, developed using the platform Limesurvey (version 3.26.0), was distributed via our private network and the crowdsourcing platform Mechanical Turk (mTurk) and was carried out in 7–14 December 2020. Overall, 60 people participated in the study, producing more than 6865 words of qualitative data which took roughly between 25 and 35 min to complete (average 28 min). The answers of our participants were balanced throughout the questions, meaning that every question had a similar response in terms of their length. However, it appeared that participants had slightly more to say regarding the questions of *Support* and *Trust*, and less about *Privacy*. Participants (32 male, 28 female) were between 23 and 71 years old with an average age of 36 years. The participants were also asked whether they have any experience with mental health issues and/or mental health-related therapies. A total of 23 participants indicated that they had experience, whereas 36 did not and one person abstained from answering. In terms of familiarity with CAs, 10 participants stated that they had never used CAs, 32 participants indicated that they were using CAs on a regular basis, and 18 participants have used CAs before, but not on a regular basis. In total, 7 of the participants using CAs stated that they were not satisfied with the usage of their CA, the rest were satisfied.

In order to analyze the data, we followed a qualitative content analysis by coding the answers of the participants, which consisted mainly of inductive category forming [41]. With the inductive formation of categories, qualitative content is examined without reference to theories, and recurring aspects and overarching concepts are recorded and designated as categories. Similar concepts are then grouped together to consequently identify all relevant concepts [41]. The coding process was independently conducted by three of the authors using the qualitative data analysis software MaxQDA. Whenever there was a discrepancy in results between the authors, a discussion was established until a consensus was reached.

*3.2. PACA Design*

In order to help the participants visualize the conversation between a patient and a PACA, we created a predefined chat record. For our simulated dialogue, we used the conversational design tool Botsociety, which allows prototyping and visualizing CAs. The conversation was provided in the form of a video and was followed by a detailed description of the PACA. Then, the participants were asked to answer a series of questions. The dialogue started with Jules (human), seeking out Raffi (PACA) to talk because something is on his/her mind. During the conversation, Jules explains that he/she is experiencing anxiety and negative thoughts again. Raffi then refers to past conversations between the two, encourages Jules, and reminds him/her of old and new affirmations they went through in the past. Towards the end of the conversation, Raffi suggests starting a meditation app for Jules. To appear supportive and trustworthy, we gave PACA an agreeable and extraverted personality. To create this personality, we used language cues that were specific

for the Big Five dimensions of agreeableness and extraversion. For example, to let Raffi appear more empathic, we used insight words (e.g., see, think) as well as agreements and compliments ("you are kind and smart") [37]. We implemented an extraverted language style by incorporating a higher verbal output, followed a "thinking out loud" writing style, and focused on pleasure talk as opposed to problem talk [37]. To make sure Raffi's language was perceived as extraverted and agreeable as possible, we undertook a twofold review: First, the conversation was refined until all authors rated the PACA's words as agreeable/extraverted. Second, we used IBM Watson's Personality Insights tool [42] to verify the personality style. The personality mining service returns percentiles for the Big Five dimensions based on the text that is being analyzed. In this context, percentiles are defined as scores that compare one person to a broader population [42]. Raffi's words received a score of 87% (Extraversion) and 73% (Agreeableness), meaning that the PACA is more extraverted than 87% of the people in the population and more agreeable than 73% of the population. The total length of the conversation lasted approximately 2:15 min. The video with the entire conversation can be viewed here: https://youtu.be/-sfSNJwCCI0 (accessed on 27 February 2022).

### 4. Results: Benefits and Caveats of PACAs in Mental Health Care

Overall, we coded three categories with seven subcategories and assigned 410 text segments to the code system. The first category *PACA Support* is divided into three subcategories: *Merits* elaborate on advantages of the PACA support in mental health care, *Demerits* illustrate the concerns the participants had with a PACA in this specific context, and *Limited Merits*, includes all the statements of those respondents who found the support of a PACA only partially helpful. The second category *PACA Trust* includes statements about the extent to which participants would trust a PACA and whether they would build a relationship over a longer period with the CA. The two codes that were derived for this category were *Trustworthy* and *Untrustworthy*. The third and final category, *PACA Privacy*, was specifically about data privacy and whether the participants would allow access to their data in order for the CA to be personality adaptive. Its two subcategories are called *Uncritical* and *Critical*. Figure 1 provides an overview of the categories and their subcategories.

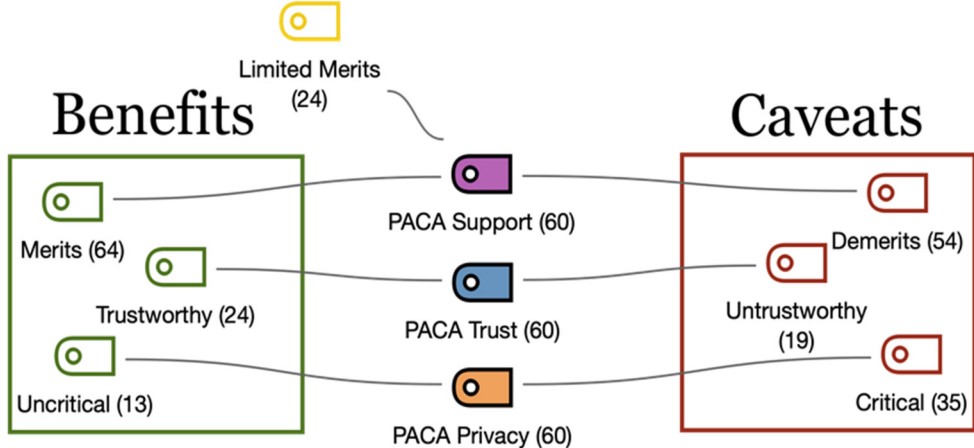

**Figure 1.** Coding System of the qualitative study.

### 4.1. PACA Support

PACA Support contains the most responses, as this category consists of three specific questions altogether (see Table 1). Support or social support includes mechanisms and activities involving interpersonal relationships to protect and help people in their daily lives [6]. One of the most mentioned demerits by the participants is that the PACA has limited skills and that this lack of ability can lead to wrong judgments. For example, one user stated: "Potential misinterpretations from the PACA of what I said could lead to more

negative thoughts and make things worse". Similarly, two other participants mention: "I would be afraid that not all of the details will be understood correctly, and maybe 'wrong' judgement will come up" and "[ . . . ] a PACA is not human and cannot fully understand the full range of issues a person is dealing with". The non-humanness of the PACA is an issue that was brought up by the participants on multiple occasions. They felt that a human therapist was necessary for mental healthcare and could not imagine interacting with a PACA. Many participants did not have a specific reason not to choose PACA or they believed that overall people are better at helping than PACAs. They expressed sentiments such as "No, I most likely will always choose a real human therapist" or "People really need an actual human to human interaction in life".

Another demerit that was mentioned several times was the mental health care context in which the CA was used for. Participants indicated that a PACA might not be supportive when it specifically comes to severe cases of complex mental health issues. They indicated that it is "probably too hard to solve for today's AI solutions" in this context. One person elaborated that "in mental health, they [PACAs] could do serious damage just by not understanding and addressing user needs". According to the participants' responses, one of the main reasons for such unforeseeable outcomes was "negative or destructive behavior" that a PACA can evoke in patients. Specifically, an "aggressive or dominant behavior" by the PACA might lead to the patient "completely closing off and losing hope". In contrast, other responses mentioned desocialization as a caveat, noting that patients can become "dependent on the PACA and start to distance from reality and real people". Other demerits stated by the participants were that communicating with a PACA would be "creepy" and "odd" for them.

One of the most frequently stated merits mentioned by the participants was the accessibility/availability of the PACA. While one participant thought the PACA "provides an escape from challenging emotional situations whenever necessary. [ . . . ] Raffi can be available when the therapist is not", another subject stated that "it can be helpful because it functions like a relief cushion for the patient while they wait for a therapist assigned to them. They feel understood and listened to, no matter how trivial the conversation with the PACA may be". Being listened to is another merit that was brought up several times. The respondents indicated the PACA would be like a "penpal" or "like the best friend you have at home". Further benefits that were listed several times include PACA's ability to put patients at ease ("it can give out apps to help soothe the mind"), to memorize past conversations ("[ . . . ] it does not forget what has already been discussed and is not annoyed when the same topic comes up again and again") and to create a personalized experience ("[ . . . ] it makes you feel like there is some more meaning to sharing it with something that can at least pretend to care. It can help personalize your experience. This makes people feel worthy").

A large proportion of the participants stated that a PACA might be helpful for mental health therapy by motivating and/or advising the patients, specifically by being a "helpful support in everyday life". One participant further pointed out that if "developed carefully, and deployed and monitored safely, PACAs have enormous potential for helping bridge the gap between patients' needs and the mental health system's available qualified staff and resources". Some respondents noted that "[ . . . ] if it feels genuine with good and varied tips/tasks/advice" and "[ . . . ] if the AI is so genuine that it's hard to distinguish from a human" they can imagine using the PACA as a support system for mental health issues. While some participants stated that they do not fear that PACA could become manipulative or pose a danger, other respondents wrote that they only partially believe in the merits of the PACA. They specifically noted that a PACA can be considered as a "short-term supporting system" that "prepares the patients mentally" but that human therapists should "regularly intervene and supervise". In fact, the suggestion that PACA should be monitored by human therapists was mentioned by majority of participants. Another limited merit that was brought up several times by the respondents is that the benefit of using a PACA largely depends on how the PACA is designed/skilled. They made comments such as

"communication style, body language and tone of voice is very important and powerful elements of communication and have a great impact on others".

*4.2. PACA Trust*

Another important factor identified in the participants' responses was trust. Trust is commonly understood as the "willingness of a party to be vulnerable to the actions of another party based on the expectation that the other will perform a particular action important to the trustor, irrespective of the ability to monitor or control that other party" [43] (p. 712). Participants mentioned that "trust is very important for all the time" and an important precondition for the development and maintenance of a long-term relationship with the PACA. However, the participants had different views on whether a PACA can build up enough trust to establish a long-term relationship, even after seeing a "real" human therapist.

On one hand, some participants argued that they would stop using the PACA when a human therapist was available. For example, one participant stated, "if I were seeing a real human therapist, I would not see the need to continue chatting with the PACA". In many cases, concerns were connected to the compatibility of two therapies at a time. That is, if the advice of the PACA were not to be aligned with the human therapist, it could cause problems for the patient. The assumption, that a human therapist always has the advantage over a PACA is leading to the decision of many participants to decide against a long-term relationship with the PACA. One respondent would see human supervision as an obligatory requirement to use a PACA over a longer period: "[...] not as long as there is no human supervision behind Raffi. Even human therapists in training have mandatory supervision".

Trust issues were also associated with the difference between humans and AI in that "a PACA is not human and cannot fully understand the full range of issues a person is dealing with". Only when it would be "hard to distinguish from a human [...] or gives such good advice", it could be a partner in a long-term relationship. For example, one participant expected the PACA to get "rid of any flaws and be very helpful in my everyday life for me to talk to it like a spouse I married". Finally, some participants expressed their general privacy concerns which would hinder any form of a long-term relationship ("I don't trust any listening device, the privacy risks are simply too great"). Especially the storage of data and the implementation of the software would need to be transparent and understandable. For example, participants stated that "I could see myself building trust only if the program continued to be trustworthy to use". Some replies connect the long-term abilities of a PACA with the quality level of the software, stating for example: "Only when the AI is so genuine that it's hard to distinguish from a human I would have contact". Accordingly, there were misgivings about the current state of AI solutions. The participants seem to believe that a human-AI relationship would only make sense if the quality level of the AI advice offered surplus benefits on top of the human therapist.

In terms of creating a bond with humans, many participants stated that they would highly appreciate abilities to memorize past conversations: "Listen, remember, build suggestions and recommendations on previous conversations not on general conclusion." Many participants seemed to define this characteristic as that of a "good listener". This desire to have an active listener in a therapy bot underlines participants' wish for individualization of therapy. This aspect is strongly correlated with the learning abilities of the PACA. To build a bond, PACA should be able to learn about the patient over time and adapt to him/her accordingly. Statements included for example: "[...] the PACA [is] required to [...] build on the previous experiences has with the person. It has to show an understanding for the situation of that person and be able to react properly." The answer also shows the importance of empathy in the conversation with a PACA. The ability to fully understand the patients' situation is only possible by learning over time and developing contextual 'thinking'. Adequate reactions of the PACA are strongly connected to its ability to assess situations or feelings the patient might have experienced.

With regard to trust building with a PACA, answers included transparency about the software's limitations. For some respondents, this would lower the concerns about security significantly. Suggestions also included assessing and evaluating the software regularly. This could include external assessments by experts or authorities as well as user feedback, that would help to improve the software.

Other than these caveats, participants were open towards establishing trust and maintaining a relationship with the PACA over a longer period. They appeared to feel "comfortable talking to it even after seeing a human therapist". One of the reasons stated was that they would "find it easier instead of constantly calling the therapist", particularly because they believe that some of their issues as "just too small to bother someone with". These participants could not only imagine themselves trusting the PACA because it can "give something back and seems to care", but also because it would be "like a friend you have at home". Moreover, participants stated that "the more you interact with it the easier it could be" to build trust and maintain a relationship with the PACA.

*4.3. PACA Privacy*

Category of PACA Privacy captured all concerns of the participants involving the necessity of a PACA to gather and analyze (sensitive) data to assess the personality of a user and adapt accordingly. Privacy refers to the non-public area in which a person pursues the free development of his or her personality undisturbed by external influences. Concerns arise when such private information could enter the public domain without authorization [44]. The participants addressed aspects that they considered to be particularly critical and the ones that they did not consider to be critical. The most important critical aspects were the potential invasion of privacy, as participants did not feel comfortable sharing personal information and "feel a little bit under a microscope". One participant stated that it "sounds alarming to allow a PACA access to your personal data and communications", while another participant said that as a "user you always need to be aware of what the information could be used for and vulnerabilities always exist". Another participant stated: "No, that's invasion of my privacy. I do not feel comfortable with allowing access to my personal messages and phone history. "Caveats against sharing personal information seem to be connected to data privacy issues on social media indicating that "the overall trust in todays messenger systems has suffered a lot over the past years due to several events." It appears that the underlying problem is the loss of control over personal data. Even if the program itself is trustworthy, the potential risk of hacking and exposing vulnerable data to criminals is reason enough not to share private chat history with a PACA.

Apart from being unwilling to give the PACA access to private data, responses also expressed concerns about the practicality from a legal perspective. Specifically, one respondent mentioned the US HIPAA laws, which regulate and provide guidance for the proper uses and disclosures of private health information. It further defines how to secure the data and what to do in case of a breach of rules [45].

The second concern was the skeptical view on data security and the possibility of data leaks. Participants were concerned if "information would be leaked or stolen" or if the system will be hacked or will malfunction. In addition, the company running a PACA needs to be trusted and a clear policy needs to be created. To support privacy, many participants mentioned that they would appreciate transparency regarding where the data is stored and processed. A PACA should offer the same privacy securities as a human therapist would do. Generally, privacy could be increased by using multiple authentication methods, such as one-time- passwords, keys, or facial recognition. Suggestions also included a decentralized data structure instead of uploading all the data into a cloud. This would mean to store data on the devices and therefore giving the user more control of his own data. To prevent the misuse of data as much as possible, participants suggested that responsibility for storing the data could be given to companies specialized in securing highly sensitive data. As the processing of data would most likely be hard to realize without cloud computing, the use of end-to-end encryption was proposed. To ensure that the processed data cannot be matched

with individuals, the software could use generated codes instead of the individuals' names. The list for matching codes and individuals could be stored separately. It was further mentioned that the privacy policy should also be certified by official institutions, which would also help to assure legal foundation of the program.

Concerning data handling, one participant suggested conducting personality tests with the patient instead of requiring full access to social media accounts. Generally, data is not supposed to be stored longer than needed. The program should also be transparent about how the data is being deleted.

On the bright side, the participants seemed to agree that they need to provide data in order for the PACA to work properly. "Yes, I would allow it to access my data. I would be willing to trust it if it could help me in the long run" said one participant. To "get the best results", the participants agreed on providing data to the PACA so that it can adapt to a user and "help my therapy in a positive way". Even though many participants had concerns about the use of sensitive data, they appeared to be willing to share their data under certain conditions to take advantage of the PACA. It seemed to be a majority opinion that data should be sent and stored in encrypted form and not passed on to third parties. They further agreed that only the critical information should be used and the data should be deleted when it was no longer needed. If these specific conditions were met and were explicitly communicated by the PACA, a disclosure of private information was acceptable by the participants. The perceived benefit from the use of the personal information was also to be communicated by the PACA and be visible to the user. Therefore, the design of the PACA and its handling of personal data is critical. Table 2 summarizes the results for all categories.

**Table 2.** Summary of the generated codes.

| Category | Subcategory | Code |
|---|---|---|
| Support | Merits | Accessibility and availability, friend/penpal, easier to talk to, puts patient at ease, memorizes conversations, personalized experience, helpful for mental health therapy, gives advice, motivates patient, good alternative to human, no judgement, and no fear of manipulation |
| | Demerits | Limited skills, wrong judgement, human therapist necessary, not helpful for severe cases, encouraging negative behavior, creepy and fake, generic answers, dependency, and desocializing |
| | Limited Merits | Prepares patients mentally, depends on personal preference, short-term support, PACA monitored by human, and depends on design and skill |
| Trust | Trustworthy | Building relationship imaginable and continuous learning about the user |
| | Untrustworthy | Long-term interaction not needed and privacy concerns as inhibitors to build rapport |
| Privacy | Uncritical | Allowing access to data, allowing access if PACA is beneficial, and transparency is important |
| | Critical | Data security, leak information, invasion of privacy, and misuse of data |

## 5. Addressing the Caveats of PACAs in Mental Health Care

The results of the survey show, that a significant number of caveats are associated with the PACAs' abilities to substitute a human and mirror the skill set of a real therapist. Comments included doubts about the PACA being able to detect severe mental illness, being able to understand the full range of problems, or be able to think contextually. While certain functionalities suggested could guarantee that the PACA is not used for cases

beyond its scope, others might be hard to accomplish with the current technological know-how. Even though AI is being constantly improved and the effectiveness of CAs has been evidenced in multiple types of research, highly complex conversation or even therapy on the level of a human therapist are not likely to be realized in the near future [7,46].

In contrast to its technological design, privacy concerns can be addressed with the current state of technology. Major caveats concerning storage and processing of data can potentially be addressed by giving data privacy a high priority during the development phase of a PACA. Encryption, decentralized data structures, and protection by multiple authentication methods are valuable suggestions, that would also help to increase trust and lower concerns. Measures to secure sensitive data, such as health-related information in cloud environments are already existing and could be implemented [46,47]. Limitations may exist in the efforts to run software completely on the local devices. Especially when it comes to privacy-related caveats, communication and transparency are very important as trust is based on data security.

Other than data privacy-based trust, the results also underline the importance of the PACAs communication style. Generally, a friendly, cheerful, but confident appearance was appreciated by the respondents. This observation aligns with the latest research on the personalities of CAs in mental health care [7]. Despite the fact, that the survey data provides insights that help to answer the initial research questions, there are further limitations to be considered. The participants of the survey were randomly chosen. Even though the survey is balanced from a demographic perspective, most of the respondents were not familiar with the concepts of PACAs and relied on the impression given in the introduction of the survey. The expertise of the users regarding how to design PACA and regarding the technological options that exist were also limited. Nevertheless, the results are sufficient to derive requirements to answer the research questions posed in this paper.

### 5.1. Requirements

One of the major concerns mentioned in the survey was the overall privacy of data. Accordingly, the security of the software system should have the highest priority. This includes the storage of data and processing of data that is necessary for the functionality of PACAs. To prevent misuse of highly sensitive data by external parties, adequate protection against attacks from hackers is required. To assure its legal correctness, usage of chat history and social media insights need to be checked or certified by the authorities. Data privacy laws are highly specific and can vary significantly between the different states [48]. Overall, the need for the users' data must be kept as low as possible.

Regarding the technological abilities of the software, it is important that the program functions flawlessly as mentally ill patients are highly sensitive towards mistreatment or undetected imminent safety risks [3]. Certification by mental health care experts or mental health authorities must be undertaken to ensure high quality care. However, it is necessary that companies offering PACAs for mental health care are transparent about their expectations, limitations, and field of use before granting full access to the PACAs services. Transparency was also considered as one of the main factors for building trust in the program. This included its functionality as well as privacy of data.

Further requirements for PACAs can be identified in the way they express themselves. In terms of verbal language, the wish for a friendly, considerate, and cheerful, but also confident and polite was mentioned several times. Balancing between cheerfulness and humor and confidence without pressuring a patient is one of the most challenging tasks for future PACAs. Body language, such as gentle, human-like gestures are expected to support the impression of the PACA.

Regarding verbal and para-verbal language cues, PACA should be required to learn about a patient the same way a human therapist would do to develop a contextual form of understanding and built long-term bond.

Finally, the PACA should be compatible with real therapy. Most participants do not see PACAs today as a substitution for therapists, which raises the question of how

both can be combined. Anyways, it must be assured, that the PACA is not promoting counterproductive advice and is the best case even involved in the therapy.

*5.2. Solution Approaches*

The following suggestions for the design of future PACAs were derived from the survey data and are aimed at reducing people's caveats for PACAs in mental health care. As overall privacy and data security were some of the main identified requirements to overcome these concerns, several actions could be considered for the future design of PACAs. To reduce the risk of misuse or leakage of patients' sensitive health data, it is recommended that a decentralized data structure is created for data storage. Wherever possible, data should be stored on the device of the user, and be protected behind its firewalls. Even though this reduces the risk of digital theft of data, physical theft or loss of the phone/computer needs to be considered as well [49]. It may be helpful to include emergency software functions that assure the deletion of the on-device data in case of loss. Similar functions are available on latest smartphone, such as Apple's iOS devices. Additionally, local data could be secured using multiple authentication methods. This could include, for example, one on time passcode (OTP) or facial recognition in addition to traditional passwords.

While the short and even long-term storage of data on the device is relatively easy to achieve, the on-device processing of data is significantly harder to accomplish. Comprehensive AI solutions, which are necessary for the realization of a PACA, usually rely on cloud computing [50]. However, the software could be run on the devices whenever the computational power is sufficient and rely on cloud computing only when strictly necessary. Further options include end-to-end encryption of data or anonymized processing of data. This would assure, that the processed information could not be tracked back to the individual user. In any case, it is recommended to run the service on infrastructure providers who specialize in the protection of sensitive data. To guarantee compliance with local data security regulations, systems should be checked in cooperation with authorities or experts.

Regarding the concerns about the conversational data, participants also expressed doubts about the necessity of granting full access to social media accounts and chat history. To reduce the amount of data that is used to adapt the PACA to the user's personality, it might be helpful to let the users decide which data they want to share. In addition, personality tests or pre-therapy conversations can be conducted with the PACA to gain information about the patients' personality traits. The usage of personality tests could also be helpful to identify severe mental illnesses or behavior that potentially exceeds the capabilities of the PACA. It would address the concern of many respondents, that PACA could possibly give inadequate advice resulting in counterproductive effects for the patients' mental health state. In addition, functions could be implemented that help people to seek proper help from therapists [3]. As many respondents see PACAs as an addition to real therapy or as a bridging function, it is recommended to develop approaches to implement PACAs into the later therapy. This could include, for example, possibilities to monitor the patients' conversations with the PACA by the therapist or reflect them during the therapy sessions. During the development process of a PACA, independent psychology experts could also examine its functionality, quality, and compatibility with therapy. Regular checks could include testing its ability to use verbal, para-verbal, and body language adequately and therefore meet patients' needs.

Finally, transparency and communication play key roles in the use of PACAs in mental health care. Especially transparency of the data security concepts and functionalities could help to reduce users' concerns. Nevertheless, if the users do not know where their data is being stored and processed, the caveats remain despite highly competitive security concepts. Transparent communication can therefore be seen as a tool to reduce people's concerns. It would further help to clearly address the abilities, limits, and scope of PACAs

to create realistic expectations and prevent misuse or disappointment. Ultimately, this is the only possibility to create long-term trust in new solutions as PACAs.

## 6. Discussion

Reflecting on the results of this study, it is important to look at how the initial motivation—globally rising numbers of mental illnesses, as well as the effects of the COVID-19 pandemic—strengthen the need for additional health care workers worldwide. In low-come countries without fully functional health care systems as in high-income countries, the situation is even worse. The number of approximately 2 health care workers per 100,000 inhabitants highlights this urgency. Against this background, CAs offer the potentials to cover or reduce the lack of health care workers. By qualitatively surveying a total of 60 people, we were able to identify potential benefits and caveats, which we then translated into general requirements and proposed solutions. The results of our study shed light on both the negative and positive aspects of PACAs and contribute to theory and practice.

### 6.1. Theoretical Contributions

As expected, most participants were more critical as opposed to not being critical towards their sensitive data. Although several participants stated that they could imagine building a trustworthy relationship with the PACA, it should not be ignored that almost as many indicated they did not find the PACA from the example trustworthy. This suggests that people perceive CAs differently and have varied preferences concerning the communication style of a CA, highlighting once again the individual differences of people. Corresponding to findings from previous studies [14] a PACA may offer helpful support to people in need, put them at ease, and can be a friend who listens when human therapists are not available—specifically in light of the pandemic, this can be considered as an enormous benefit. However, in line with existing research [3,11], PACAs may also create an unintended (emotional) dependency which, for example, can lead to further reduction in socializing. If these issues are not addressed properly, Weizenbaum's caveat of a "Nightmare Computer" could indeed come true. In the 1960's, AI capabilities were limited, and much like her namesake Eliza Doolittle from the play "Pygmalion" [51], Weizenbaum's ELIZA had no understanding of the actual conversation but merely simulated a discourse with intelligent phrasing. Yet, ELIZA simulated her psychotherapeutic conversations so convincingly that people became deeply and emotionally involved with the program. This demonstrates how "simple" verbal communication can be used or taken advantage of to achieve positive or negative outcomes. With today's powerful AI capabilities, the current critical voices regarding AI ethics are therefore very much justified. Focusing on the design of CAs without carefully considering any potential consequences for people's well-being can backfire quickly. Although humans do know from a philosophical perspective that machines are not capable of expressing "real" feelings, they still respond to them emotionally as if they are. A CA's poor communication skills and specifically that of a PACA that can personalize to the user's communication preference on a high level, could aggravate negative health outcomes instead of improving them.

Our findings contribute to the strong and nascent research stream on AI in the sustainability field. In this context, AI has been studied primarily for ethical aspects [12], for possible sustainable business practices [52], or specifically how AI in the context of the sustainable development goals [53]. In particular, the factors of sociotechnical systems that are unsustainable were highlighted. With these findings in mind, we place our research right within this research stream and show how AI can be used in the context of healthcare and specifically mental health care.

### 6.2. Practical Implications

Based on the insights from our participants, requirements were derived that would need to be fulfilled to reduce the expressed concerns of the users. Priorities for derivable

requirements include the assurance of absolute privacy of data via modern security concepts and flawlessly functioning software. Best case, both points should be confirmed by external authorities or experts. The survey data suggests that a mere fulfillment of requirements concerning data security or software development is not sufficient to reduce concerns. Even though they need to be fulfilled, the communication and transparency concerning data privacy as well as functionality and limitations of the software are equally important to be addressed. In addition, the analysis of the survey data raises questions of ethical concerns and whether they are possible to be solved in the near future. Other than the aforementioned privacy concerns and the risk of harm, the general risk of bias in machine learning algorithms also exists for PACAs. The data base that is used for training the AI is strongly correlated with its later focus on certain characteristics of the data, such as gender or ethnical origin [3]. It is therefore essential not to train algorithms that could lead to possible biases and thus reinforce discrimination or exclusion.

For implementing a PACA and offering such a service, it is therefore important to follow a set of guidelines to create a safe and benevolent service. These guidelines result from the discussed requirements and solution approaches: (1) Guarantee a high level of data security through current standards, (2) high transparency of the interaction and capabilities of a PACA, and (3) development together with experts (therapists and psychologists) and without bias.

Furthermore, it is important not to advertise such a service as a substitute for therapy, but as a first point of contact or an accompaniment to therapy. The marketing of a PACA in the field of mental health care is therefore just as important as its design and implementation. Furthermore, we advise companies to consider how PACAs can be implemented and offered. We also advise governments and non-profit organizations to consider how AI can be used to address the major problem of mental health issues. Organizations such as the WHO can learn from our findings to launch initiatives to promote AI and PACAs and to elaborate support and funding possibilities. In this case, it is necessary not only to offer commercial systems (such as Woebot or Replika), which cannot necessarily be used in low-income regions of the world, but also to implement and offer free to use and non-profit systems (PACAs in particular).

## 7. Conclusions

In our paper we show that there are existing ways to overcome peoples' concerns of PACAs in mental health care. Utilizing the technological development of AI solutions [54], PACAs open the potential of helping to reduce the worldwide gap of mental health care workers by supporting users in their daily life. Even if state-of-the-art solutions are not ready to fully substitute human therapists, prevention alone could support reducing the pressure in global mental health care systems. In this context, our findings show the potential not only of PACAs, but also of CAs, or AI more generally. However, limitations of the paper exist especially in its potential to formulate fully functional solution approaches. Suggestions made by participants during the literature review focus on how caveats can be reduced. Nevertheless, the technical feasibility remains unproven for several implications concerning the design of the software and data security concept. The practicality of these concepts is therefore beyond the scope of this paper and requires further research in the future. Another limitation is the small number of participants, which is not a representative user group. All people are affected by mental health and are therefore possible users. A representative group would therefore be important to query all benefits and caveats, as well as possible requirements. However, through our different dissemination strategies (use of the crowdsourcing platforms Mturk and distribution in the private environment), we could approach this representative user group.

In conclusion, PACAs offer a great potential to address the worldwide problem of mental health issues.

**Author Contributions:** Conceptualization, R.A. and D.S.; methodology, R.A., D.S., and H.H.; formal analysis, R.A., D.S. and H.H.; investigation, R.A. and D.S.; writing—original draft preparation, R.A. and D.S.; writing—review and editing, R.A., D.S., H.H., and T.d.V.; visualization, R.A.; supervision, R.A., D.S., and T.d.V.; project administration, R.A. and D.S. All authors have read and agreed to the published version of the manuscript.

**Funding:** We acknowledge support by the Open Access Publication Funds of the Technische Universität Braunschweig.

**Institutional Review Board Statement:** Ethical review and approval were waived for this study as our data collection did not include any personal data and no information was captured to identify our participants.

**Informed Consent Statement:** Informed consent was obtained from all subjects involved in the study.

**Data Availability Statement:** Not applicable.

**Conflicts of Interest:** The authors declare no conflict of interest.

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
