# Peer review of "Requirements and Solution Approaches to Personality-Adaptive Conversational Agents in Mental Health Care"

_sustainability, doi:10.3390/su14073832_

Round 1
Reviewer 1 Report
The article is well written, there are, however, areas the research can be improved. In the abstract, conversational agents (line 15) should be written in initial caps because they are presented as nouns and are allocated an acronym CAs. Similar also for Personality adaptive conversational agents (line 25), change to initial caps. Make similar corrections in the main text also.
I have been unable to understand the research problem, in the paragraph in line 57, the authors highlight the attributes of PACA, and at the end present a research question 1 asking about the benefits of PACA, which the authors mentioned in the preceding paragraph.
“PACA automatically infers users’ personality traits and adapts accordingly to their personality by using language that is specific to a particular personality dimension (e.g., extraversion, agreeableness), with the aim to enhance dialogue quality.”
“Transferred to “virtual psychotherapy”, PACAs may be able to establish rapport with the patient to enhance interaction quality and mental health support”.
The quoted texts between lines 57 and 65 already present the benefits of PACA, why do the authors ask about the benefits of PACA when the question has been answered by the literature? Kindly clarify
Research questions 1 and 2 are the same. Both research questions do not seem to be relevant based on the discussed literature that have provided ample validation about the benefits of PACA. You can, however, decide to keep the research question if the reason will be to validate or negate previous study on PACA, this, the authors must clearly state in their work.
In the methodology, authors should be specific when the study was carried and duration. December 2020 is vague. Was the study conducted during a one month period of December 1-31, 2020?
In the results and discussions, you must include few more citations from Scopus/WoS sources. Please consider to add the following citations and references in your academic article. Please also consider to cite the following international studies
Chaveesuk, S., Khalid, B., & Chaiyasoonthorn, W. (2019, August). Emergence of New Business Environment with Big Data and Artificial Intelligence. Proceedings of the 9th International Conference on Information Communication and Management. https://doi.org/10.1145/3357419.3357441
Chaiyasoonthorn, W., Khalid, B., & Chaveesuk, S. (2019, August). Success of Smart Cities Development with Community’s Acceptance of New Technologies. Proceedings of the 9th International Conference on Information Communication and Management. https://doi.org/10.1145/3357419.3357440
Muangmee, C. (2021). Effects of Facebook advertising on sustainable brand loyalty and growth: case of Thai start-up businesses. Transnational Corporations Review, 1–12. https://doi.org/10.1080/19186444.2021.1986340
Other sections of the study look good.
Author Response
Dear editor and reviewers,
thanks you for your valuable comments, which answer in the following reviewer response document. We identified 21 individual comments which we ordered according to the reviewers. We addressed all comments and revised our manuscript accordingly. We think that due to the reviewers comments, we were able to significantly improve the manuscript.
Best regards,
the authors

Reviewer 2 Report
Herein, the authors collect a set of relevant requirements for designing PACAs and provide solution approaches that can be followed to design and implement PACAs for mental health care.
- The main motivation of the paper should be polished in the introductory section.
- I would appreciate it, if the authors include in the introductory section, the cons/pros (if any) of the work.
- For better presentation, the authors have to include a list of abbreviations, following the journal's style in this regard.
- Extending the discussion part is necessary.
- Move the conclusion part to a new part.
- I urge the authors to add some advising info to WHO.
- Update the review literature with recent relevant work ,preferable, from MDPI journals, also, cite the following seminal textbook: Winston, Patrick Henry. Artificial intelligence. Addison-Wesley Longman Publishing Co., Inc., 1992.
Author Response

(The authors gave the same response as above.)

Reviewer 3 Report
Dear authors,
The title of your research is interesting, however, I could barely find any connection to sustainability or the "Digital Innovation and Transformation in Healthcare"! It makes the research somehow irrelevant to this special issue. Nevertheless, as I can see the potential in your manuscript, I would like to ask you to submit a revised version.
To submit a revised version, please re-write the abstract. Make your abstract attractive to readers (simple sentences without any repetition) and include 2-3 sentences ready to be cited exactly as they are. In 1 paragraph, your abstract should tell the readers why the study is important (maximum 25% of the text), what you did, i.e. your methodology (maximum 25% of the text), and what you found, i.e. main research results and their major implications (50% of the text). This is very important to promote your work because of the growing trend that authors use Google search to find and cite papers based on the abstract (instead of reading the full paper).
Besides, what is the specific research stream you have found on the Sustainability journal that can include your contribution? how does the paper push the research forward? please, be more explicit on this issue.
The first research question must be "RQ1: What are the benefits of PACAs in mental health care?"
Please add more evidence under the "2.3. Personality-Adaptive Conversational Agents" section. Many recent studies have been overlooked.
Please provide more evidence about the interview protocol.
It is mentioned that "The survey, developed using the platform Limesurvey (version 3.26.0), was distributed via our private network and the crowdsourcing platform Mechanical Turk (mTurk) and was carried out in December 2020." The role of researchers is marginalised. Why? This could be a major limitation.
Please compare your findings with those of the others.
Besides, please highlight the limitations and implications. Do you have any idea about the generalizability of the findings?
All this said, I find the thrust of your paper interesting and hope you will be able to make the revisions needed to make it publishable. I certainly appreciate your willingness to submit your work to the Sustainability journal.
Best of luck!
Author Response

(The authors gave the same response as above.)

Round 2
Reviewer 2 Report
The authors successfully addressed all comments raised in my early report and the paper can now be considered for publication.
Author Response
Dear editor and reviewers,
thank you again for providing final comments to our manuscript. We identified 4 individual comments, which mostly focus on language improvement. We furthermore slightly improved the literature and the whole manuscript underwent proofreading.
Best regards, the authors

Reviewer 3 Report
Dear authors,
Thank you for submitting a revised version. I could see you have correctly addressed most of the comments. Nevertheless, it would be best to improve the literature, especially by referring to the previously published journals in Sustainability or any other relevant journal. Besides, English proofreading is required.
I would love to see the last edition.
Best of luck!
Author Response

(The authors gave the same response as above.)
